# Density responses of lesser-studied carnivores to habitat and management strategies in southern Tanzania's Ruaha-Rungwa landscape

**Marie Hardouin**[1]*, **Charlotte E. Searle**[2], **Paolo Strampelli**[2], **Josephine Smit**[3,4], **Amy Dickman**[2], **Alex L. Lobora**[5], **J. Marcus Rowcliffe**[1,6]

**1** Faculty of Natural Sciences, Imperial College London, Ascot, United Kingdom, **2** Wildlife Conservation Research Unit, Department of Zoology, The Recanati-Kaplan Centre, Tubney, United Kingdom, **3** Southern Tanzania Elephant Program, Iringa, Tanzania, **4** Department of Psychology, University of Stirling, Stirling, United Kingdom, **5** Tanzania Wildlife Research Institute, Arusha, Tanzania, **6** Institute of Zoology, Zoological Society of London, London, United Kingdom

* marie.acx.hardouin@gmail.com

**Data Availability Statement:** The data used in this study is freely available at https://doi.org/10.5281/zenodo.4041430.

## Abstract

Compared to emblematic large carnivores, most species of the order Carnivora receive little conservation attention despite increasing anthropogenic pressure and poor understanding of their status across much of their range. We employed systematic camera trapping and spatially explicit capture-recapture modelling to estimate variation in population density of serval, striped hyaena and aardwolf across the mixed-use Ruaha-Rungwa landscape in southern Tanzania. We selected three sites representative of different habitat types, management strategies, and levels of anthropogenic pressure: Ruaha National Park's core tourist area, dominated by *Acacia-Commiphora* bushlands and thickets; the Park's miombo woodland; and the neighbouring community-run MBOMIPA Wildlife Management Area, also covered in *Acacia-Commiphora*. The Park's miombo woodlands supported a higher serval density (5.56 [Standard Error = ±2.45] individuals per 100 km$^2$) than either the core tourist area (3.45 [±1.04] individuals per 100 km$^2$) or the Wildlife Management Area (2.08 [±0.74] individuals per 100 km$^2$). Taken together, precipitation, the abundance of apex predators, and the level of anthropogenic pressure likely drive such variation. Striped hyaena were detected only in the Wildlife Management Area and at low density (1.36 [±0.50] individuals per 100 km$^2$), potentially due to the location of the surveyed sites at the edge of the species' global range, high densities of sympatric competitors, and anthropogenic edge effects. Finally, aardwolf were captured in both the Park's core tourist area and the Wildlife Management Area, with a higher density in the Wildlife Management Area (13.25 [±2.48] versus 9.19 [±1.66] individuals per 100 km$^2$), possibly as a result of lower intraguild predation and late fire outbreaks in the area surveyed. By shedding light on three understudied African carnivore species, this study highlights the importance of miombo woodland conservation and community-managed conservation, as well as the value of by-catch camera trap data to improve ecological knowledge of lesser-studied carnivores.

**Funding:** Scholarship funding for CS and PS is provided by the University of Oxford NERC Environmental Research DTP (https://www.environmental-research.ox.ac.uk). AD is funded by a Recanati-Kaplan Fellowship (https://www.wildcru.org). JMR is supported by Research England funding to the Institute of Zoology (https://re.ukri.org/funding/). Additional funding was awarded to CS for this research from National Geographic Society Early Career Grants (https://www.nationalgeographic.org/funding-opportunities/grants), Cleveland Metroparks Zoo Africa Seed Grants (https://www.clevelandmetroparks.com/zoo), Chicago Zoological Society Chicago Board of Trade (CBOT) Endangered Species Fund (https://www.czs.org/Chicago[1]Zoological-Society/Conservation%20Leadership/CBOT-Endangered-Species-Fund/), and Pittsburgh Zoo & PPG Aquarium Conservation & Sustainability Fund (https://www.pittsburghzoo.org/conservation/). The funders had no role in study design, data collection and analysis, decision to publish, or preparation of the manuscript.

**Competing interests:** The authors have declared that no competing interests exist.

## Introduction

Emblematic large carnivores often benefit from significant conservation investment and are prioritised in terms of resource allocation over more threatened species [1]. Such prioritisation stems in part from their essential role in shaping ecosystems [2], but also their charisma and cultural significance, which are used to highlight the problem of biodiversity loss to the public [3, 4]. In comparison, carnivores with smaller body size or less aesthetic appeal tend not to spark as much conservation interest and remain understudied globally [1, 5]. The African conservation landscape conspicuously illustrates these disparities, with iconic apex predators such as lion (*Panthera leo*), leopard (*Panthera pardus*) and cheetah (*Acinonyx jubatus*) drawing substantial research attention [6]. Overshadowed by these flagship species, many lesser-studied carnivores nonetheless face declining trends in number and range [7], with anthropogenic threats including habitat loss/fragmentation [8], and persecution [9] jeopardising their survival alongside a growing human population [10].

This lack of research hinders conservation status assessments for a number of African carnivores, thus precluding effective conservation planning [11], notwithstanding the ecological importance and vulnerability to extinction of many of these species. For instance, striped hyaena (*Hyaena hyaena*) have received very little conservation attention across their African range, although they provide essential ecosystem services in consuming carcasses and dispersing their nutrients [12]. The species is thought to be declining globally, mainly caused by poisoning and decreasing sources of carrion, leading to its classification as "Near Threatened" [13]. Moreover, little information exists on the population status of many mesocarnivores, despite their key contribution to regulating lower trophic levels and maintaining biodiversity [14]. Classified as "Least Concern", serval (*Leptailurus serval*) and aardwolf (*Proteles cristata*) nevertheless face increasing pressure due to continued urbanisation and agricultural expansion [15, 16]. Serval are particularly vulnerable to the degradation of wetland and grassland habitat, as the species prefers well-watered environments which attract higher densities of small mammals [17]. The insectivorous diet of the aardwolf exposes the species to insecticide poisoning, an ever-increasing threat with the intensification of agriculture [15, 18].

The IUCN recommends further research to bridge knowledge gaps for these lesser-studied carnivore species' spatial ecology and population status [13, 15, 16]. The distribution of serval, striped hyaena and aardwolf, as well as population abundances and densities across their range, need to be investigated and accurately estimated to inform effective management strategies and conservation planning [19]. The cryptic nature of these species often impedes direct observation, and camera trapping offers an efficient alternative for collecting data [20]. Over the past decade, spatial capture-recapture [21], a well-established method for estimating the density of species with individual markings such as patterned coats, has been successfully used for several populations of these three carnivore species. Most of the published estimates for striped hyaena, however, concern Asiatic populations [22–24], with a single study of an African population from Kenya [25]. Spatial capture-recapture studies of aardwolf density are similarly limited, with only two published estimates from Kenya and Botswana [25, 26]. A few more surveys have been carried out for serval in Southern Africa [26–29], and Western and Central Africa [30, 31], revealing large variations in density across the species' range. The lack of estimates and the disparity in published results suggest the need to investigate additional populations of serval, striped hyaena and aardwolf in areas where robust estimates do not exist. Such information is essential to better understand how and why population status varies across the species' geographic range [32, 33].

The second largest National Park in East Africa at over 20,000 km², Tanzania's Ruaha National Park (RNP) belongs to the greater Ruaha-Rungwa landscape, which harbours

important wildlife biodiversity, partly due to the convergence of *Acacia-Commiphora* and miombo woodland (*Brachystegia-Jubelnardia*) ecotypes [34]. The Ruaha-Rungwa landscape also encompasses a range of fully and partially protected areas, including Game Reserves, Game Controlled Areas, Open Areas, and community-managed Wildlife Management Areas (WMA). Established on communal lands, WMAs are one of the most novel features in Tanzania's conservation landscape, which intend to promote a sustainable scheme involving local communities in wildlife protection and its economic benefits [35]. Various sources and levels of anthropogenic pressure add to the vulnerability of carnivores in the Ruaha-Rungwa landscape, including: habitat conversion for agriculture, prey base depletion due to bushmeat poaching, and direct persecution by humans in retaliation to livestock predation [36, 37]. RNP also lies at the southern range limit for both the global striped hyaena population and the East & Northeast African aardwolf population [13, 15]. This ecosystem is therefore of particular conservation interest for the species, since peripheral populations often differ in density response and appear more sensitive to habitat change than more centrally located populations [38].

In addition to providing baseline status estimates for three African carnivore species for which insufficient data currently exist, this study intended to investigate local variations in density across different habitats, management strategies, and levels of protection and anthropogenic pressure. To this end, we used camera trapping and spatially explicit capture-recapture (SECR) modelling [39] to estimate population densities of serval, striped hyaena and aardwolf at three sites in Ruaha-Rungwa, representative of different ecosystem types and levels of protection and anthropogenic pressure: the *Acacia-Commiphora* core tourist area of RNP; the miombo woodland of western RNP; and the community-run *Acacia-Commiphora* MBO-MIPA WMA, which adjoins Ruaha to the east and acts as a buffer between the Park and unprotected village lands.

## Materials and methods

### Ethics statement

Data collection consisted of camera trapping, a non-invasive method which avoids contact with the study species and minimises interference with their natural behaviour. Fieldwork was carried out under research permits 2018-368-NA-2018-107, 2019-96-ER-97-20 and 2019-424-NA-2018-184, granted by the Tanzania Commission for Science and Technology (COSTECH; Dar es Salaam, Tanzania; rclearance@costech.or.tz) and Tanzania Wildlife Research Institute (TAWIRI; Arusha, Tanzania; researchclearance@tawiri.or.tz).

### Study area

The Ruaha-Rungwa ecosystem in southern Tanzania extends over 45,000 km$^2$ across three ecoregions: Central Zambezian miombo woodlands, Eastern miombo woodlands, and Eastern African acacia savannas [40] (Fig 1A and 1B). RNP lies at the centre of this ecosystem and was the largest National Park in Tanzania until the formation of Nyerere National Park in 2019 [41]. With an altitude ranging from 716 m to 1,888 m [42], the terrain features low rolling hills overlooked by an escarpment to the north, and the Great Ruaha River runs along the RNP's eastern border [43]. RNP experiences a semi-arid to arid climate and a unimodal rainfall regime, from December to April [44]. *Acacia-Commiphora* deciduous bushlands and thickets predominate in about two-thirds of the park, while miombo woodlands cover its western part [45] (Fig 1B). A number of Game Reserves, Game Controlled Areas, and Open Areas complete the Ruaha-Rungwa landscape to the north and the west of RNP, while the Matumizi Bora ya Malihai Idodi na Pawaga (MBOMIPA) and Waga WMAs stretch along its eastern border.

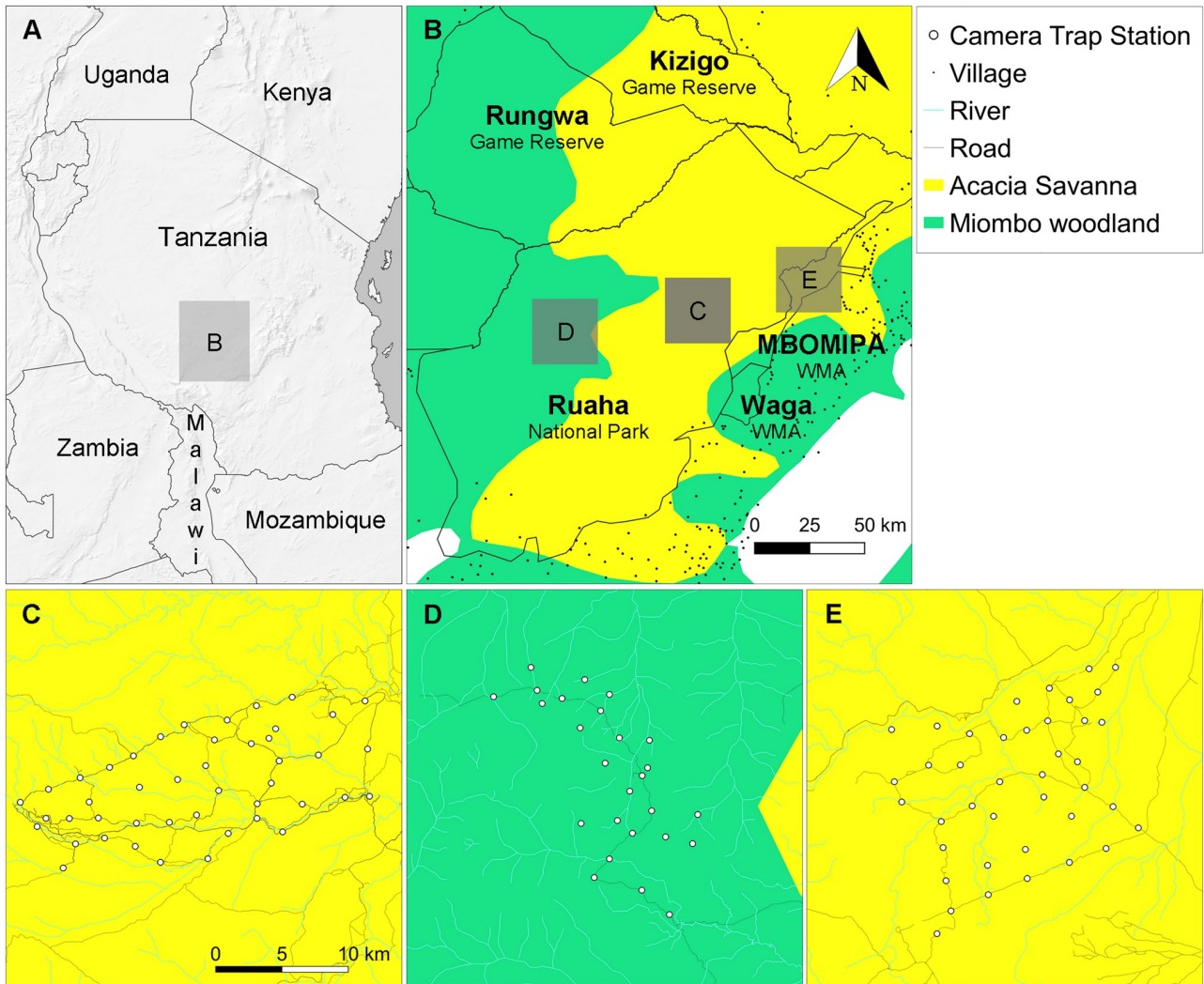

**Fig 1. Ruaha-Rungwa landscape and spatial distribution of camera trap stations.** (A) Location of the Ruaha-Rungwa landscape in Tanzania (made with Natural Earth). (B) Ruaha-Rungwa landscape's ecotypes [40] and land uses. The map depicts, but does not explicitly name, boundaries of additional protected areas, and only shows villages and towns near protected areas. (C) Core RNP *Acacia-Commiphora* grid. (D) RNP miombo grid. (E) MBOMIPA WMA *Acacia-Commiphora* grid.

MBOMIPA WMA was established in 2007 based on the principles of community-based natural resource management [35] and encounters greater levels of anthropogenic impacts than the core tourist area of RNP [46]. The landscape is unfenced, which facilitates free movement of wildlife.

## Camera trap surveys

The study used a photographic dataset collected between June and November 2018 in RNP and MBOMIPA WMA to assess the status of leopard [46]. We surveyed two sites within RNP, and one in MBOMIPA WMA (Fig 1C–1E). The first camera trap grid covered an area of 223 km$^2$ in highly-productive riverine *Acacia-Commiphora* habitat [47], at the core of the park's tourist area, with 45 camera stations set out for 83 days. The second grid, located in miombo woodland in the western part of RNP, consisted of 26 camera stations over an area of 152 km$^2$

for 90 days. We deployed the third grid in highly-productive riverine *Acacia-Commiphora* habitat in MBOMIPA WMA, with 40 stations spanning a total area of 270 km$^2$ for 69 days.

The duration of each survey was kept below 90 days both to ensure sufficient captures/ recaptures and to approximate closed populations, as per previous studies [20, 21, 28, 48–52]. The design of the camera trap layout sought to optimise capture rates of large carnivores, facilitate individual identification, and ensure no gaps in the trapping array. Camera placement prioritised roads and junctions, or game trails, as large carnivores preferentially travel along roads [53]. With a single road crossing the study area within the park's miombo woodland, half of the cameras in the RNP miombo grid were set on game trails along semi-open *mbugas* (drainage lines). The mean distance between two cameras ranged from 1.88 km in RNP miombo to 1.96 km and 2.08 km in Core RNP *Acacia-Commiphora* and WMA *Acacia-Commiphora*, respectively. Given the published home range estimates for the target species [54, 55], this spacing enabled to recapture individuals at multiple camera trap stations [21, 56], in compliance with SECR requirements for modelling space use.

All but one of the stations consisted of paired cameras facing one another on either side of a road or trail, to increase the likelihood of capturing both flanks of passing animals [48]. Cameras were mounted on trees at approximately 40 cm height, with the surrounding vegetation regularly cleared to prevent any obstruction of the lens or sensor and reduce the risk of bush fire damaging the camera. The survey deployed several motion-activated camera models: Cuddeback Professional Color Model 1347 and Cuddeback X-Change Color Model 1279, Non Typical Inc., Wisconsin, USA; HC500 HyperFire, Reconyx, Wisconsin, USA. The majority of cameras featured xenon flashes, which produce higher definition images of animal markings than infrared LED flashes [56].

## Density estimation

We estimated the population density (defined as the number of individuals per 100 km$^2$) of serval, striped hyaena, and aardwolf at each survey site via Maximum Likelihood Estimation SECR analysis, with the package secr v3.2.0 [57] in R version 3.6.3 [58]. SECR combines two sub-models representing the spatial distribution of a population and the detection process respectively [59], and therefore does not require ad hoc estimation of the effective trapping area as per conventional capture-recapture methods [60].

Data inputs consisted of detection histories detailing sampling occasions and locations of captures for each individual throughout the survey, and a trap layout listing the coordinates of every camera trap station and their activity periods (S2 and S3 Appendices). We visually identified individuals in camera trap images through their unique pelage markings [48], focusing on flanks for serval and fore-quarters and hind-quarters for striped hyaena and aardwolf, as detailed in S1 Appendix. We sexed individuals based on the unobstructed view of external genitalia, late pregnancy signs such as weight gain and enlarged abdomen (see S1 Appendix), or the presence of cubs. Individuals whose sex could not be confidently distinguished were classified as "unknown sex" (coded NA in the detection histories). We selected the flank with the greatest number of capture events for each species and grid to produce detection histories, using the following framework: (i) sampling occasions spanned 24-hours from midday to midday, to account for the nocturnal nature of the studied species; (ii) individuals could be recorded at different locations during a sampling occasion, but only once per given location (as per standard practice [61]).

The SECR observation sub-model describes the individual detection probability as a monotonous function of the distance to home range centre, characterised by g0, the detection probability at the range centre, and σ, a spatial scale related to home range width [62]. We

tested three standard functions (half-normal, negative exponential and hazard rate) to model the decrease of the detection function with the distance to the home range centre. To minimise any bias in estimated densities, we selected a buffer width at which density estimates reached a plateau for each analysis.

We tested for variation in the baseline encounter probability g0 through the use of embedded predictors and covariates hypothesised to influence detection probability [63]. Specifically, we investigated the behavioural response of individuals to capture events by fitting the global trap response model (b predictor) and the local trap response model (bk predictor), and assessed the influence of the type of flash at each station (xenon versus LED flash), and station location (on-road versus off-road) on detection. Finally, we examined the effect of sex on detection function parameters g0 and σ by fitting a hybrid mixture model, which accommodates individuals of unknown sex [64]. Model selection was carried out by ranking models based on their Akaike Information Criterion score, corrected for small sample size (AICc) [65]. When more than one candidate model had substantial empirical support (ΔAICc < 2) [65], a final density estimate was derived by applying model averaging based on AICc weights [21].

## Results and discussion

### Results

**Survey effort.**    The sampling effort across the 111 stations deployed at the three study sites totalled 8,477 camera trap nights and yielded 226 images of serval, 125 images of striped hyaena and 1,119 images of aardwolf (see S4 Appendix). We could identify individuals in 86.3% of serval photographs, 72.8% of striped hyaena photographs, and 81.1% of aardwolf photographs. Detection histories for serval featured 38 independent capture events for 13 identified individuals in Core RNP *Acacia-Commiphora*, 31 events for 10 identified individuals in WMA *Acacia-Commiphora*, and 23 events for 12 identified individuals in RNP miombo. Striped hyaena were captured in WMA *Acacia-Commiphora* only, with 42 independent capture events and 12 identified individuals. Aardwolf were only captured at camera stations in Core RNP *Acacia-Commiphora* and WMA *Acacia-Commiphora*, with 36 and 37 individuals identified, accounting for a total of 240 and 185 independent capture events, respectively.

Aardwolf displayed the highest recapture rate, i.e. the percentage of individuals captured more than once, across all grids (84.9%), followed by striped hyaena (58.3%) and serval (57.1%). All captured individuals were adults or subadults, with the exception of one female serval and one aardwolf accompanied by offspring. The target species showed predominantly nocturnal activity, with 86% of identified serval captures, 98% of striped hyaena captures, and 99.4% of aardwolf captures occurring between 7 pm and 6 am. The poor visibility of genitalia and lack of clear secondary sexual traits limited sex determination to 2.9% of the identified servals and 16.7% of identified striped hyaenas, preventing the modelling of sex differences in the density estimation process. Aardwolf sexing proved more successful, with sex assigned to 35.6% of identified individuals, which allowed us to model sex differences in detection function parameters.

**Population density.**    Of the three detection functions tested, the negative exponential function best fitted the datasets for all target species at the three sites. We found no evidence of learned behavioural response or influence of covariates on detection for serval in Core RNP *Acacia-Commiphora* and WMA *Acacia-Commiphora*. However, the model in which g0 varied with station location was supported in RNP miombo. S5 Appendix details the ranking results based on AICc score. RNP miombo yielded the highest serval density, with 5.56 [Standard Error = ±2.45] individuals per 100 km$^2$, followed by 3.45 [±1.04] individuals per 100 km$^2$ in

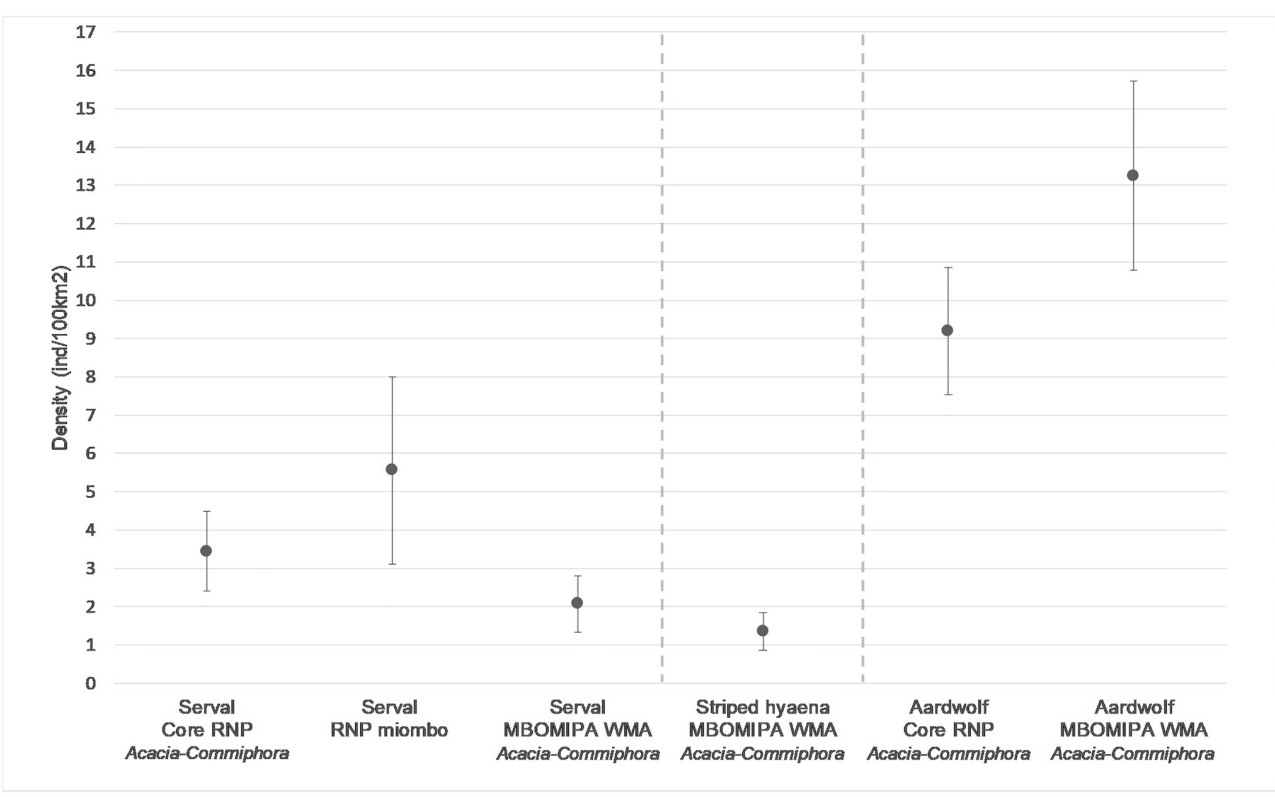

**Fig 2. Population density estimates for the target species at the three survey sites.** Bars represent standard errors.

Core RNP *Acacia-Commiphora*, and 2.08 [±0.74] individuals per 100 km² in WMA *Acacia-Commiphora* (Fig 2, Table 1).

For striped hyaena in WMA *Acacia-Commiphora*, two models received strong support (ΔAICc < 2, [65]); the model with constant g0 and sigma, and the local trap response model

**Table 1. Population density and parameter estimates, with standard errors/95% confidence intervals for the target species at the three survey sites.**

| | | | Estimate ± SE | | Estimate ± SE 95% CI |
|---|---|---|---|---|---|
| **Species** | **Site** | **Predictor** | **g0** | **σ(m)** | **Density (ind/100 km²)** |
| Serval | Core RNP *Acacia-Commiphora* | | 0.041 ± 0.014 | 1062 ± 171 | 3.45 ± 1.04 |
| | | | | | 1.93–6.16 |
| | RNP miombo | On road | 0.033 ± 0.017 | 1249 ± 361 | 5.56 ± 2.45 |
| | | Off road | 0.006 ± 0.004 | | 2.44–12.69 |
| | WMA *Acacia-Commiphora* | | 0.034 ± 0.013 | 1566 ± 299 | 2.08 ± 0.74 |
| | | | | | 1.05–4.09 |
| Striped hyaena | WMA *Acacia-Commiphora* | bk = 0 | 0.032 ± 0.016 | 2625 ± 521 | 1.36 ± 0.50 |
| | | bk = 1 | 0.078 ± 0.029 | | 0.68–2.72 |
| Aardwolf | Core RNP *Acacia-Commiphora* | bk = 0 | 0.030 ± 0.009 | 1220 ± 131 | 9.19 ± 1.66 |
| | | bk = 1 | 0.263 ± 0.038 | | 6.46–13.06 |
| | WMA *Acacia-Commiphora* | Female | 0.135 ± 0.085 | 476 ± 111 | 13.25 ± 2.48 |
| | | Male | 0.323 ± 0.075 | 553 ± 42 | 9.22–19.05 |

Predictor bk- = 1 for site-specific step change after the first capture.

(bk). Model averaging returned an estimated density of 1.36 [±0.50] individuals per 100 km$^2$.

For aardwolf in Core RNP *Acacia-Commiphora*, learned behavioural response influenced capture probability, with the local trap response model (bk) ranking highest, while the models in which g0 or sigma varied with sex ranked highest in WMA *Acacia-Commiphora*. Aardwolf density was estimated at 9.19 [±1.66] individuals per 100 km$^2$ in Core RNP *Acacia-Commiphora*, and 13.25 [±2.48] individuals per 100 km$^2$ in WMA *Acacia-Commiphora*.

## Discussion

**Density comparison.** This study produced the first population densities estimated through spatial capture-recapture for three lesser-studied carnivores, serval, striped hyaena and aardwolf, in Tanzania. Using camera trap data from the Ruaha-Rungwa landscape, we shed light on the population density responses of these increasingly threatened species to habitat and land management strategies. Our results provide an important baseline and reference estimate that can be employed for future ecological monitoring of striped hyaena, aardwolf and serval.

Serval densities were higher within the RNP grids than in WMA *Acacia-Commiphora*, and in particular within the miombo woodland. This density distribution follows the precipitation pattern over the Ruaha-Rungwa landscape, as the miombo woodlands in the south-western part of RNP receive the greatest amount of rainfall on average and WMA *Acacia-Commiphora* is the driest of the three surveyed areas [44]. The miombo woodlands feature open, grassy *mbuga* ("dambo") drainage lines. Seasonal rains flood mbugas and water remains present year-round, usually in distinct springs or pools during the dry season. This result aligns with the preference of servals for well-watered environments such as wetlands and riparian habitat [17]. The difference between RNP miombo and Core RNP *Acacia-Commiphora* could also stem from the lower abundance of lion, leopard and spotted hyaena (*Crocuta crocuta*) in the miombo woodlands [46], as these apex predator species occasionally kill serval [66]. The lower serval population density in WMA *Acacia-Commiphora* likely results from the WMA's proximity to unprotected land, inducing greater edge effects [67], and a more pronounced level of anthropogenic pressure. For instance, the camera stations in the WMA recorded nine illegal incursions, two of them with evidence of bushmeat hunting, as well as two spotted hyaenas and two striped hyaenas bearing marks of snare captures. The common use of dogs for bushmeat hunting may also negatively impact resident populations of serval, as occasional killings by domestic dogs have been observed elsewhere in Tanzania [68]. In contrast, the survey only recorded one illegal incursion in the RNP grids. Moreover, the lower abundance of small ungulates in the WMA driven by anthropogenic impacts and the lack of habitat immediately outside the WMA may intensify competition between serval and leopard [47]. In areas with fewer impala or other medium-sized antelopes, leopard may shift their diet towards smaller prey species, such as large rodents and birds, and thus increase their dietary overlap with serval [69].

Serval density estimates using spatial capture-recapture vary greatly across Africa, from 0.63 and 1.28 individuals per 100 km$^2$ in northern Namibia [28] to 101.21 in Mpumalanga, South Africa [29]. The latter estimate comes from a heavily modified habitat where the absence of persecution and interspecific competition, combined with high rodent biomass, enables serval to attain such high numbers. In comparison, our results lie within the lower range of published densities and are close to estimates for the Niokolo Koba National Park, Senegal (3.49–4.08 individuals per 100 km$^2$) [31] and the Drakensberg Midlands, South Africa (6.0–8.3 individuals per 100 km$^2$) [27]. The only other density estimate available for the species in Tanzania

exceeds our results, with 10.9 [SE = 3.17] individuals per 100 km² in Tarangire National Park [70]; however, the study employed conventional capture-recapture, which has been shown to overestimate density compared to spatial capture-recapture [71].

We could only estimate striped hyaena population density in WMA *Acacia-Commiphora*, as the camera trap survey did not detect any individuals in Core RNP *Acacia-Commiphora* or RNP miombo. The Ruaha-Rungwa landscape lies at the southern limit of the species' global range [13]; deeper within the species' range, in Laikipia County, Kenya, population density estimated through SECR was around six times higher than the density estimated in WMA *Acacia-Commiphora* [25]. Such a pattern is consistent with widespread biogeographic trends, typically showing diminished abundance towards species range edges [72]. Furthermore, the species is closely associated with *Acacia-Commiphora* bushland within Africa [68], and the fact that our study landscape traverses a transition from this habitat to miombo woodland may point to a mechanism delimiting the range edge in this area. The low density observed in WMA *Acacia-Commiphora* may also ensue from high edge effects due to the area's proximity to human-dominated areas [67], as highlighted by the record of two individuals showing evidence of wire snare capture during the study period.

In contrast to serval, we estimated a higher aardwolf density in WMA *Acacia-Commiphora* than in Core RNP *Acacia-Commiphora*. The difference probably results from milder intraguild predation in the WMA, which supports lower densities of apex predators than the core tourist area of RNP [46]. Moreover, as aardwolf primarily feed on harvester termites, depletion of ungulates would not trigger dietary overlap with larger carnivores [18]. In addition, aardwolf may benefit from accidental fire outbreaks which often occur later in the dry season in the WMA; the reduced grass cover caused by this burning may force harvester termites to range more widely in search of food, thus lengthening their exposure time to predation on the ground [73, 74]. The reverse trend in aardwolf density compared to serval could indicate different sensitivities to anthropogenic pressures between the two species, although poaching activities in the area do not specifically target either species. Aardwolf may also be more prone to predation by apex predators, with greater stealth and agility giving serval an edge on the slower-moving aardwolf when threatened [17]. The absence of records of aardwolf in miombo woodland reflects the species' known preference for *Acacia–Commiphora* bushland [68]. Our results align with aardwolf population densities estimated through conventional capture-recapture and spatial capture-recapture in Tarangire National Park and Kenya's Laikipia District, of 9.04 and 11.63 individuals per 100 km², respectively [25, 70].

Camera trap location impacted detection probability for serval in RNP miombo. Rather than denoting serval's preference for larger roads, the higher detection probability at stations located on roads probably results from the particular road network in this portion of RNP, consisting of a single road complemented by two infrequently-used vehicle tracks and numerous subsidiary game trails. Though we tried to select the most heavily used trails, their large number offers wildlife multiple equivalent options compared to the stations set on the most readily accessible road. For aardwolf in Core RNP *Acacia-Commiphora* and striped hyaena, the probability of detection at a camera trap station was higher for individuals that had already been captured at the same site. This suggests that individuals favour the core area of their home range more often than the periphery and are not deterred by camera trap flashlights (xenon or LED flash). Sex appeared to have a mild impact on aardwolf capture probability in WMA *Acacia-Commiphora*, with males having a higher probability of capture than females.

**Conservation implications and recommendations.** This study made use of photographic by-catch data from a research project targeting leopard. Considering the logistical and budget

requirements of camera trapping, this study illustrates the value of by-catch data in terms of resource optimisation, provided that the survey design also complies with the ecology of the non-target species [75]. Our trap layout met spatial sampling recommendations for the species we considered: the array coverage areas were larger than single home range areas of all species [54, 55, 76, 77], and spacing was sufficient to yield recaptures of individuals across multiple stations in all cases [21, 76, 78–80]. However, the trap layout configuration was tailored to maximise leopard capture rate, with stations preferentially deployed on roads and junctions [53], which may not be optimum for serval, striped hyaena and aardwolf. For instance, striped hyaena have been recorded to move mostly cross-country rather than along roads in the Serengeti [12]. Similarly, mesocarnivores such as serval and aardwolf may avoid areas commonly used by apex predators, which would lower their capture rate along roads [81]. Although the analysis did not find any evidence of road avoidance behaviour, we suggest that future work targeting these species might improve detectability by positioning camera stations around sightings of the animals or their signs.

The higher population density estimated for serval in miombo woodland indicates that this is an important habitat for the species in the region. One of the most extensive ecosystems in Sub-Saharan Africa, miombo woodlands cover a substantial portion of Tanzania's protected and unprotected land [82]. Miombo plays a key role in the livelihood of rural communities but faces unsustainable resource exploitation [83, 84]. As such, increased woodland degradation and deforestation would negatively impact serval as well as a range of other mammal species. As miombo woodlands have received little conservation attention compared to *Acacia* habitats in East Africa [82], we recommend additional survey work across their extent to improve the understanding of these systems and protect their unique biodiversity.

Additionally, our results highlight the importance of the community-managed MBOMIPA WMA for the striped hyaena and aardwolf in the Ruaha-Rungwa landscape. Along with the adjacent RNP, MBOMIPA WMA provides the two hyaenids with suitable habitat at the edge of their global range. More generally, MBOMIPA WMA acts as a buffer zone between the highly protected RNP and the surrounding villages, and our results demonstrate the potential of community conservation in complementing more traditional conservation strategies, particularly in areas with lower densities of vulnerable species such as striped hyaena. Nevertheless, the camera trap survey found evidence of bushmeat hunting in the WMA; although pictures show that hyaenas can break free from snares and survive with amputated limbs, snaring and other non-selective poaching methods generally induce an increase in anthropogenic mortality of non-target species in savannah ecosystems [85, 86]. Considering the species' low density in the Ruaha-Rungwa landscape, a conservation priority for striped hyaena should aim to reduce the impact of snaring on resident populations and assess the corresponding risk to their viability.

In conclusion, our study illustrates how data from camera trap surveys targeting charismatic large carnivores can shed light on populations of lesser-studied carnivores, broadening the scope of capture-recapture studies beyond focal species. While adjustments to sample design can help to extend the range of accessible species in surveys [26], in this case, we were able to generate useful data without such adjustments. Similar efforts, employing by-catch data, could be easily replicated elsewhere to provide information on the spatial and population ecology of a range of species. Finally, we recommend complementing the baseline estimates provided for serval, striped hyaena and aardwolf in the Ruaha-Rungwa landscape with investigations into spatial variation in site use across the landscape to determine the environmental and anthropogenic factors driving their habitat selection.

## Supporting information

**S1 Appendix. Individual identification and sexing.**
(PDF)

**S2 Appendix. Capture histories.**
(PDF)

**S3 Appendix. Trap layouts.**
(PDF)

**S4 Appendix. Survey grid summary information.**
(PDF)

**S5 Appendix. SECR model ranking.**
(PDF)

## Acknowledgments

We would like to thank the Government of Tanzania, Tanzania Wildlife Research Institute (TAWIRI), Tanzania Commission for Science and Technology (COSTECH), Tanzania National Parks Authority (TANAPA), Tanzania Wildlife Management Authority (TAWA), and Idodi-Pawaga MBOMIPA WMA. We also thank the field staff of the Southern Tanzania Elephant Program (STEP) and Ruaha Carnivore Project (RCP) for their assistance with data collection, all WMA Village Game Scouts, TANAPA Rangers, and TAWA Game Scouts who contributed to fieldwork.

## Author Contributions

**Conceptualization:** Marie Hardouin, Charlotte E. Searle, Paolo Strampelli, J. Marcus Rowcliffe.

**Data curation:** Charlotte E. Searle.

**Formal analysis:** Marie Hardouin.

**Funding acquisition:** Charlotte E. Searle, Paolo Strampelli, Josephine Smit, Amy Dickman.

**Investigation:** Charlotte E. Searle, Paolo Strampelli, Josephine Smit.

**Methodology:** Marie Hardouin.

**Supervision:** Charlotte E. Searle, Paolo Strampelli, J. Marcus Rowcliffe.

**Writing – original draft:** Marie Hardouin.

**Writing – review & editing:** Marie Hardouin, Charlotte E. Searle, Paolo Strampelli, Josephine Smit, Amy Dickman, Alex L. Lobora, J. Marcus Rowcliffe.

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
