## [Decision Letter · Decision Letter 0]

26 Nov 2020

PONE-D-20-33931

Density responses of lesser-studied carnivores to habitat and management strategies in southern Tanzania’s Ruaha-Rungwa landscape

PLOS ONE

Dear Dr. Hardouin,

Thank you for submitting your manuscript to PLOS ONE. After careful consideration, we feel that it has merit but does not fully meet PLOS ONE’s publication criteria as it currently stands. Therefore, we invite you to submit a revised version of the manuscript that addresses the points raised during the review process.

We look forward to receiving your revised manuscript.

Kind regards,

Bi-Song Yue, Ph.D

Academic Editor

PLOS ONE

"We would like to thank the Government of Tanzania, Tanzania Wildlife Research

 Institute (TAWIRI), Commission for Science and Technology (COSTECH), Tanzania National

Parks Authority (TANAPA), Tanzania Wildlife Management Authority (TAWA), and Idodi

Pawaga MBOMIPA WMA for their support of this research."

"Scholarship funding for CS and PS is provided by the University of Oxford NERC Environmental Research DTP (https://www.environmental-research.ox.ac.uk). AD is funded by a Recanati-Kaplan Fellowship (https://www.wildcru.org). Additional funding was awarded to CS for this research from National Geographic Society Early Career Grants (https://www.nationalgeographic.org/funding-opportunities/grants), Cleveland Metroparks Zoo Africa Seed Grants (https://www.clevelandmetroparks.com/zoo), Chicago Zoological Society Chicago Board of Trade (CBOT) Endangered Species Fund (https://www.czs.org/Chicago-Zoological-Society/Conservation-Leadership/CBOT-Endangered-Species-Fund), and Pittsburgh Zoo & PPG Aquarium Conservation & Sustainability Fund (https://www.pittsburghzoo.org/conservation/). The funders had no role in study design, data collection and analysis, decision to publish, or preparation of the manuscript."

4. We note that Figure 1 in your submission contain map images which may be copyrighted. All PLOS content is published under the Creative Commons Attribution License (CC BY 4.0), which means that the manuscript, images, and Supporting Information files will be freely available online, and any third party is permitted to access, download, copy, distribute, and use these materials in any way, even commercially, with proper attribution. For these reasons, we cannot publish previously copyrighted maps or satellite images created using proprietary data, such as Google software (Google Maps, Street View, and Earth). For more information, see our copyright guidelines: http://journals.plos.org/plosone/s/licenses-and-copyright.

4.1.    You may seek permission from the original copyright holder of Figure 1 to publish the content specifically under the CC BY 4.0 license. 

4.2.    If you are unable to obtain permission from the original copyright holder to publish these figures under the CC BY 4.0 license or if the copyright holder’s requirements are incompatible with the CC BY 4.0 license, please either i) remove the figure or ii) supply a replacement figure that complies with the CC BY 4.0 license. Please check copyright information on all replacement figures and update the figure caption with source information. If applicable, please specify in the figure caption text when a figure is similar but not identical to the original image and is therefore for illustrative purposes only.

Reviewers' comments:

Reviewer's Responses to Questions

**Comments to the Author**

1. Is the manuscript technically sound, and do the data support the conclusions?

Reviewer #1: Yes

Reviewer #2: Partly

Reviewer #3: Yes

2. Has the statistical analysis been performed appropriately and rigorously? 

Reviewer #1: Yes

Reviewer #2: No

Reviewer #3: Yes

3. Have the authors made all data underlying the findings in their manuscript fully available?

Reviewer #1: Yes

Reviewer #2: Yes

Reviewer #3: Yes

4. Is the manuscript presented in an intelligible fashion and written in standard English?

Reviewer #1: No

Reviewer #2: Yes

Reviewer #3: Yes

5. Review Comments to the Author

Reviewer #1: Hardouin et al.

This paper is better than it might appear. As a comparison of three sites, with no replicates, one might dismiss it as a pedestrian. As an ecological study, it is. What makes it important is its methods. The authors use bycatch — and I just love their use of that term — the incidental records of species in a study of leopards. There are now surely thousands of camera trap studies aimed are finding large mammals that, often enough, have other species walk by. These other species are interesting, important, and sometimes poorly known. They are grist to this paper’s mill and good for the authors.

Second, the methods of setting up the trapping grid and identifying the individuals also provide a model for other studies. I see a lot of camera trap studies to review: this is a good one.

Oh, but the English is awful. I suspect the senior author is not a native speaker, and so has an excuse. The other authors need to be ashamed. I have rarely seen so many sentences written in the passive voice.

Overall, a basic, unpretentious paper that is useful contribution to a genre that will assume importance with the expanding number of papers about camera trapping.

Reviewer #2: Specific Comments:

Lines 21-22: Is ‘less charismatic’ necessary? Also, the use of body size adjectives is relative. What is large and what is smaller? The species listed could easily be viewed as large carnivores in many systems.

Lines 31-33: It isn't clear here what the 'respectively' is referencing. Is it the 3 sites listed above in the order they are listed?

Line 46: The use of camera traps is not novel. There are many, many published studies that have used this technique. I wouldn’t suggest it as a rationale or novelty of the study here.

Lines 85-91: Is this difference in geography important? How so? Is the distinction of just these 2 countries important? Do you expect differences compared to other parts of the geographic range for this species? Justification for study seems tenuous and not convincing given how it is presented.

Lines 109-110: The technique is not novel. Based on the rationale here and the issue of geographically incomplete data sets, the need for study is questionable, as presented.

Lines 155-156: So no movement in or out over 90 days. This doesn't seem realistic.

Lines 161-163: What is the significance here on distance? Was independence of camera trap being sought?

Lines 180-189: So the entire 90 days was extent of the capture history matrix? How wouldn't there be deaths, immigration and emigration occurring across such a long time period? Thus, if this population is actually open, why not use a Jolly-Seber model to assess population numbers? How certain was the individual identification? Any estimate of your error rate?

Lines 205-206: Any concern with count data and overdispersion of data?

Lines 228-232: I don't recall any mention of diagnostics used to identify gender in the Methods. Any biases and how were they minimized?

Lines 243-245: AIC weights range from 0 to 1 but the values that are presented in S5 Appendix range from 0 to >71. The delta AICs are not calculated correctly either, at least given the AICc values presented. Did the authors interpret results based on the values presented? If so, I would question the results and interpretations. Also, if models were similar in terms of the AIC weight, were the parameters and their SE evaluated to see if a model had uninformative variables (sensu Arnold 2010. Journal of Wildlife Management)?

Lines 343: I don't understand the point of this first sentence.

Lines 345-348: How so and what does this mean to be 'compatible with the home range'?

Lines 354-356: Does this conclusion follow based directly on the design of the study and analyses? I didn't see this as an objective in the Introduction.

Reviewer #3: This paper used photos of three mesocarnivores, aardwolf, serval, and striped hyena, from a previous camera trap study of leopards along with spatial capture-recapture modeling methods to estimate their population abundance/density in Tanzania. The authors set up the need for the study well by explaining the conservation challenges for these medium-sized carnivores in that they are all facing a variety of threats but do not receive the same conservation attention as charismatic megafauna (like lions). Because the three species involved have patterned coats with unique markings among individuals, the authors developed capture histories for use in the modeling (included in the supporting information). Using SECR, they estimated population density for each of the three species while addressing model assumptions and testing the potential influence of the camera flash and station location on detection. Their results provide density estimates for the three species and are given in context to habitat, previous density estimates, and conservation concerns. Overall, I thought the paper was really well written and the spatial capture-recapture modeling was thorough and addressed the important assumptions involved in using this approach. The density estimates all appear realistic given the species and habitat and were effectively compared to other published results. I also appreciated the discussion of the data in the context of conservation.

There are two points for clarification that would strengthen the paper. First, it would be helpful to add details to the methods for identification of individual animals using the pelage patterns. Correctly identifying unique individuals is essential for their use of capture-recapture analysis. When describing their identification methods, the authors cite a paper from 1995 that manually identified 10 individual tigers so I assume that is the method they also used but it is unclear. The supplemental materials include images of each species with polygons around markings but there’s no caption to explain whether this was done by an observer or software like Hotspotter. It would be helpful to include more of a description about how the individuals were identified and ID was confirmed.

Secondly, the authors note that that large carnivores preferentially use roads for movement and the camera photos used were originally collected for a study of leopards. However, there is no mention of the potential intraspecific interactions between the three focal species in this study and leopard (or even interactions among the three species). Was this considered as a covariate in the model and if so, why was it not included? Adding addition information and discussion about the intraspecific responses would be helpful.

6. PLOS authors have the option to publish the peer review history of their article (what does this mean?). If published, this will include your full peer review and any attached files.

Reviewer #1: No

Reviewer #2: No

Reviewer #3: No

---

## [Author Response · Author response to Decision Letter 0]

10 Mar 2021

Response:

We renamed the supporting information files in accordance with PLOS ONE guidelines, i.e. “S1_File.pdf”, and we uploaded them individually.

Response:

Tanzania Commission for Science and Technology (COSTECH) and Tanzania Wildlife Research Institute (TAWIRI) are the full names of the institutions which granted research permits and approved field site access. We updated the paragraph with their headquarters location and research clearance email address, as well as an additional permit number, as follows (lines 126-129): “Fieldwork was carried out under research permits 2018-368-NA-2018-107, 2019-96-ER-97-20 and 2019-424-NA-2018-184, granted by the Tanzania Commission for Science and Technology (COSTECH; Dar es Salaam, Tanzania; rclearance@costech.or.tz) and Tanzania Wildlife Research Institute (TAWIRI; Arusha, Tanzania; researchclearance@tawiri.or.tz).”

3. Thank you for stating the following in the Acknowledgments Section of your manuscript: "We would like to thank the Government of Tanzania, Tanzania Wildlife Research Institute (TAWIRI), Commission for Science and Technology (COSTECH), Tanzania National Parks Authority (TANAPA), Tanzania Wildlife Management Authority (TAWA), and Idodi Pawaga MBOMIPA WMA for their support of this research."

Please remove any funding-related text from the manuscript and let us know how you would like to update your Funding Statement. Currently, your Funding Statement reads as follows: "Scholarship funding for CS and PS is provided by the University of Oxford NERC Environmental Research DTP (https://www.environmental-research.ox.ac.uk). AD is funded by a Recanati-Kaplan Fellowship (https://www.wildcru.org). Additional funding was awarded to CS for this research from National Geographic Society Early Career Grants (https://www.nationalgeographic.org/funding-opportunities/grants), Cleveland Metroparks Zoo Africa Seed Grants (https://www.clevelandmetroparks.com/zoo), Chicago Zoological Society Chicago Board of Trade (CBOT) Endangered Species Fund (https://www.czs.org/Chicago-Zoological-Society/Conservation-Leadership/CBOT-Endangered-Species-Fund), and Pittsburgh Zoo & PPG Aquarium Conservation & Sustainability Fund (https://www.pittsburghzoo.org/conservation/). The funders had no role in study design, data collection and analysis, decision to publish, or preparation of the manuscript."

Response:

The institutions mentioned in the Acknowledgments Section did not provide funding for this research but administrative, logistics, and intellectual support. Therefore, our Funding Statement does not need to include them. We amended the Acknowledgments Section to avoid potential confusion, as follows (lines 409-412): “We would like to thank the Government of Tanzania, Tanzania Wildlife Research Institute (TAWIRI), Tanzania Commission for Science and Technology (COSTECH), Tanzania National Parks Authority (TANAPA), Tanzania Wildlife Management Authority (TAWA), and Idodi-Pawaga MBOMIPA WMA."

4. We note that Figure 1 in your submission contain map images which may be copyrighted. All PLOS content is published under the Creative Commons Attribution License (CC BY 4.0), which means that the manuscript, images, and Supporting Information files will be freely available online, and any third party is permitted to access, download, copy, distribute, and use these materials in any way, even commercially, with proper attribution. For these reasons, we cannot publish previously copyrighted maps or satellite images created using proprietary data, such as Google software (Google Maps, Street View, and Earth). For more information, see our copyright guidelines: http://journals.plos.org/plosone/s/licenses-and-copyright.

4.1. You may seek permission from the original copyright holder of Figure 1 to publish the content specifically under the CC BY 4.0 license. We recommend that you contact the original copyright holder with the Content Permission Form (http://journals.plos.org/plosone/s/file?id=7c09/content-permission-form.pdf) and the following text: “I request permission for the open-access journal PLOS ONE to publish XXX under the Creative Commons Attribution License (CCAL) CC BY 4.0 (http://creativecommons.org/licenses/by/4.0/). Please be aware that this license allows unrestricted use and distribution, even commercially, by third parties. Please reply and provide explicit written permission to publish XXX under a CC BY license and complete the attached form.”

Please upload the completed Content Permission Form or other proof of granted permissions as an "Other" file with your submission. In the figure caption of the copyrighted figure, please include the following text: “Reprinted from [ref] under a CC BY license, with permission from [name of publisher], original copyright [original copyright year].”

4.2. If you are unable to obtain permission from the original copyright holder to publish these figures under the CC BY 4.0 license or if the copyright holder’s requirements are incompatible with the CC BY 4.0 license, please either i) remove the figure or ii) supply a replacement figure that complies with the CC BY 4.0 license. Please check copyright information on all replacement figures and update the figure caption with source information. If applicable, please specify in the figure caption text when a figure is similar but not identical to the original image and is therefore for illustrative purposes only.

Response:

In Figure 1A, we used for the shaded relief a shapefile from the Natural Earth website (https://www.naturalearthdata.com). As declared on the Natural Earth website, “All versions of Natural Earth raster + vector map data found on this website are in the public domain”. Ecoregions in Figures 1B-E originate from the WWF website (https://www.worldwildlife.org/publications/terrestrial-ecoregions-of-the-world), which did not mention any copyright issues. Other elements of the maps such as country’s borders, protected areas delineation, rivers, roads, or human settlements also originate from shapefiles which are not copyrighted. We mentioned the Natural Earth website and the reference for ecoregions in Figure 1 caption, as follows (lines 147-152):

“Fig 1. Ruaha-Rungwa landscape and spatial distribution of camera trap stations. (A) Location of the Ruaha-Rungwa landscape in Tanzania (made with Natural Earth). (B) Ruaha-Rungwa landscape’s ecotypes [40] and land uses. The map depicts, but does not explicitly name, boundaries of additional protected areas, and only shows villages and towns near protected areas. (C) Core RNP Acacia-Commiphora grid. (D) RNP miombo grid. (E) MBOMIPA WMA Acacia-Commiphora grid.”

 

Review Comments to the Author

Reviewer #1:

Hardouin et al.

This paper is better than it might appear. As a comparison of three sites, with no replicates, one might dismiss it as a pedestrian. As an ecological study, it is. What makes it important is its methods. The authors use bycatch — and I just love their use of that term — the incidental records of species in a study of leopards. There are now surely thousands of camera trap studies aimed are finding large mammals that, often enough, have other species walk by. These other species are interesting, important, and sometimes poorly known. They are grist to this paper’s mill and good for the authors.

Second, the methods of setting up the trapping grid and identifying the individuals also provide a model for other studies. I see a lot of camera trap studies to review: this is a good one.

Oh, but the English is awful. I suspect the senior author is not a native speaker, and so has an excuse. The other authors need to be ashamed. I have rarely seen so many sentences written in the passive voice.

Overall, a basic, unpretentious paper that is useful contribution to a genre that will assume importance with the expanding number of papers about camera trapping.

Response:

The manuscript was edited and proofread by native English speakers to improve the language standard and to reduce the use of passive voice.

 

Reviewer #2:

Specific Comments:

Lines 21-22: Is ‘less charismatic’ necessary? Also, the use of body size adjectives is relative. What is large and what is smaller? The species listed could easily be viewed as large carnivores in many systems.

Response:

We deleted the references to species’ charisma and body size and amended the manuscript as follows (lines 21-23): “Compared to emblematic large carnivores, most species of the order Carnivora receive little conservation attention despite increasing anthropogenic pressure and poor understanding of their status across much of their range.”

Lines 31-33: It isn't clear here what the 'respectively' is referencing. Is it the 3 sites listed above in the order they are listed?

Response:

We rephrased the sentence for clarification, as follows (lines 31-33): “The Park’s miombo woodlands supported a higher serval density (5.56 [Standard Error = ±2.45] individuals per 100 km2) than either the core tourist area (3.45 [±1.04] individuals per 100 km2) or the Wildlife Management Area (2.08 [±0.74] individuals per 100 km2).”

Line 46: The use of camera traps is not novel. There are many, many published studies that have used this technique. I wouldn’t suggest it as a rationale or novelty of the study here.

Response:

The manuscript was amended as follows (lines 44-47): “By shedding light on three understudied African carnivore species, this study highlights the importance of miombo woodland conservation and community-managed conservation, as well as the value of by-catch camera trap data to improve ecological knowledge of lesser-studied carnivores.”

Lines 85-91: Is this difference in geography important? How so? Is the distinction of just these 2 countries important? Do you expect differences compared to other parts of the geographic range for this species? Justification for study seems tenuous and not convincing given how it is presented.

Lines 109-110: The technique is not novel. Based on the rationale here and the issue of geographically incomplete data sets, the need for study is questionable, as presented.

Response:

We have replied to these two comments simultaneously as they both relate to the justification of the study. Population densities vary across species’ geographical range in response to environmental and anthropogenic factors [1]. For instance, published estimates for serval density using spatial capture-recapture show large variations depending on the ecosystem considered, from 0.63 individuals per 100 km2 in northern Namibia [2] to 101.21 in Mpumalanga, South Africa [3]. In light of such variability, large-scale extrapolations would be ill-advised, leading us to state the need to investigate populations in areas where robust estimates do not exist, such as East Africa. More generally, assessing density across different parts of a species’ range, particularly across different habitats and land use types, and surveying different components of a landscape is important to get a better understanding of how population status varies across the species’ geographic extent [4,5].

In addition, the Ruaha-Rungwa landscape presents interesting characteristics for the study of our target species. First, it lies at the southern limit of the striped hyaena geographic range and the East & Northeast African aardwolf population range. As a result of the biogeographic pattern of density decline toward the boundaries of a species range, peripheral populations are expected to be less abundant and more sensitive to habitat change than more centrally located populations [6]. Therefore, we hypothesised that striped hyaena and aardwolf densities in the Rungwa-Ruaha landscape would differ from the estimates in Laikipia County, Kenya, to which we refer in the manuscript [7]. Second, the Ruaha-Rungwa landscape’s variety of ecotypes and management strategies described in the introduction allows to study local variations in density and assess the species’ response against environmental and anthropogenic factors. We amended the introduction (lines 88-94 and 106-115) to include this additional information.

Lines 161-163: What is the significance here on distance? Was independence of camera trap being sought?

Response:

Spatial Capture-Recapture (SCR) uses location-specific individual encounter histories to construct a spatial model of the detection process, where the individual detection probability is characterised by the detection probability at the range centre, and a spatial scale related to home range width [8]. To estimate this spatial scale parameter, SCR models need some individuals be re-encountered at several camera trap sites, which constitutes an additional level of information compared to conventional capture–recapture models. Therefore, the distance between camera stations should allow the observation of individuals at multiple camera stations [9-11]. According to published home range estimates [12,13], the station spacing adopted in the survey design was compatible with spatial recaptures for serval, striped hyaena and aardwolf, despite the survey primarily targeting leopard, a species associated with larger home ranges. We added a sentence at the end of the paragraph to clarify this point (lines 171-173).

Lines 155-156: So no movement in or out over 90 days. This doesn't seem realistic.

Lines 180-189: So the entire 90 days was extent of the capture history matrix? How wouldn't there be deaths, immigration and emigration occurring across such a long time period? Thus, if this population is actually open, why not use a Jolly-Seber model to assess population numbers? How certain was the individual identification? Any estimate of your error rate?

Response:

We acknowledge that the sampling periods used in our study, i.e. 70, 83 and 90 days, increase the risk of violating the population closure assumption and introducing a bias in density estimations. However, we also note that shorter sampling periods cannot guarantee closed populations and also produce a bias [9]. Considering the low detectability of the studied species, the lengthening of the sampling period allowed for a larger number of individuals and recapture events, resulting in more accurate density estimations [9,14]. Even so, the number of serval and striped hyaena individuals in our study remained less than 15, as shown in S4 Appendix. Such bias/precision trade-off frequently occurs for elusive species, with sampling periods of 60-90 days widely adopted as a compromise between ensuring sufficient data and approximating closed populations [2,9,14,15,16,17]. Moreover, a study of the bias-precision trade-off showed a positive impact of longer sampling periods for slow-life history species and intermediate-life history species (up to 3 months) [15]. Lines 162-164 were amended to introduce this notion of compromise when choosing sampling durations.

The lead author performed individual identification twice and identifications were verified by a different observer to minimise mismatches. Any photographs with uncertain identification were excluded from the analysis. However, based on the verification performed by the second observer, we estimate that the average error rates for serval and hyaena aligned with the results of a recent snow leopard study [18] i.e. circa 10%. For aardwolf we anticipate a slightly higher error rate, between 15-20%, as aardwolf markings are less numerous and distinctive than the other study species. We added more details about individual identification into S1 Appendix.

Lines 205-206: Any concern with count data and overdispersion of data?

Response:

Overdispersion in SCR models most likely arises from non-independent spatial configurations between individuals such as clustering into groups [19]. Amongst the studied species, serval are solitary and the two hyaenids display solitary foraging behaviour despite forming monogamous pairs for aardwolf or resting in small groups for striped hyaena [20]. The only case where more than one individual featured in a camera trap photograph corresponded to females with dependant young, and we did not include accompanying offspring in detection histories. We tested the goodness of fit of the supported models with a Monte-Carlo test using the residual deviance divided by the residual degrees of freedom method [21], and the results did not show any evidence of a lack of fit.

Lines 228-232: I don't recall any mention of diagnostics used to identify gender in the Methods. Any biases and how were they minimized?

Response:

Sexing was based on the unobstructed view of external genitalia, late pregnancy signs such as weight gain and enlarged abdomen, or the presence of cubs. Individuals whose sex could not be confidently distinguished were classified as “unknown sex” (coded NA in the detection histories). We added a sentence in the manuscript (lines 195-198) and some pictures in S1 Appendix to clarify this point.

Lines 243-245: AIC weights range from 0 to 1 but the values that are presented in S5 Appendix range from 0 to >71. The delta AICs are not calculated correctly either, at least given the AICc values presented. Did the authors interpret results based on the values presented? If so, I would question the results and interpretations. Also, if models were similar in terms of the AIC weight, were the parameters and their SE evaluated to see if a model had uninformative variables (sensu Arnold 2010. Journal of Wildlife Management)?

Response:

We made a mistake when reporting data in S5 Appendix, with the last three columns of the table corresponding to {AIC, AICc, �AICc} instead of {AICc, �AICc, AICcwt}. We apologise for this error and amended S5 Appendix with the correct values. The case of a model falling within 2 AIC units (�AIC < 2) of the top-ranked model occurred twice, for striped hyaena and aardwolf in MBOMIPA WMA Acacia-Commiphora. Both aardwolf models (g0[sex] σ[.] and g0[.] σ[sex]) have 5 parameters, whereas the second-ranked model for striped hyaena (g0[.] σ[.]) has one parameter less than the top model (g0[bk] σ[.]). Therefore, �AICc between the top and second ranked models mainly result from differences in the maximised log-likelihood of the model and not from the addition of an uninformative parameter [22,23].

Lines 343: I don't understand the point of this first sentence.

Response:

We amended this sentence in the manuscript to link it more clearly to the following one about the value of by-catch data, as follows (lines 359-360): “This study made use of photographic by-catch data from a research project targeting leopard.”

Lines 345-348: How so and what does this mean to be 'compatible with the home range'?

Response:

Simulation studies investigating the impact of spatial design on SCR parameter estimates have shown that trap arrays should cover at least one home range for SCR models to perform well [24,25]. Based on the maximum distances between spatial recaptures (see S4 Appendix) and the published home range estimates for our study species [12,13], our survey design complied with this requirement. Simulations have also demonstrated the impact of trap spacing on model precision, with the existence of an optimal spacing range between 1–3 σ under the half-normal encounter probability model with spatial scale parameter σ [9,24,26,27,28]. Our survey design also met this requirement, with σ exceeding half the distance between traps for all species and sites except aardwolf in WMA Acacia-Commiphora. In any case, the design allowed recaptures of individuals across multiple stations We added a sentence in the manuscript to detail these two points (lines 362-366).

Lines 354-356: Does this conclusion follow based directly on the design of the study and analyses? I didn't see this as an objective in the Introduction.

Response:

Based on observations of the species’ behaviour in published studies, we highlight in this paragraph a potential limitation of the survey design which may have reduced capture rates and make recommendations for future work to prevent it. Our analysis tested the influence of station location on detection probability and did not find any evidence of road avoidance behaviour. However, the positioning of off-road camera stations prioritised areas used by leopards, which might have also resulted in a lower capture rate for mesocarnivores. Therefore, further information about the species’ movement in the Ruaha-Rungwa ecosystem would be needed to state confidently on this topic and future work should ideally investigate the species’ preferences prior to camera trapping. We rephrase the sentence in the manuscript for more clarity (lines 371-374). 

Reviewer #3:

This paper used photos of three mesocarnivores, aardwolf, serval, and striped hyena, from a previous camera trap study of leopards along with spatial capture-recapture modeling methods to estimate their population abundance/density in Tanzania. The authors set up the need for the study well by explaining the conservation challenges for these medium-sized carnivores in that they are all facing a variety of threats but do not receive the same conservation attention as charismatic megafauna (like lions). Because the three species involved have patterned coats with unique markings among individuals, the authors developed capture histories for use in the modeling (included in the supporting information). Using SECR, they estimated population density for each of the three species while addressing model assumptions and testing the potential influence of the camera flash and station location on detection. Their results provide density estimates for the three species and are given in context to habitat, previous density estimates, and conservation concerns. Overall, I thought the paper was really well written and the spatial capture-recapture modeling was thorough and addressed the important assumptions involved in using this approach. The density estimates all appear realistic given the species and habitat and were effectively compared to other published results. I also appreciated the discussion of the data in the context of conservation.

There are two points for clarification that would strengthen the paper. First, it would be helpful to add details to the methods for identification of individual animals using the pelage patterns. Correctly identifying unique individuals is essential for their use of capture-recapture analysis. When describing their identification methods, the authors cite a paper from 1995 that manually identified 10 individual tigers so I assume that is the method they also used but it is unclear. The supplemental materials include images of each species with polygons around markings but there’s no caption to explain whether this was done by an observer or software like Hotspotter. It would be helpful to include more of a description about how the individuals were identified and ID was confirmed.

Response:

Individual identification was performed by visually inspecting coat markings. The lead author examined and named each record once, then carried out a second run to check each of them. A different observer subsequently verified identifications to minimise mismatches, and any photographs with uncertain identification were excluded from the analysis. We also tested the I3S pattern software on a sample batch of serval pictures, but visual identification proved more effective. We started with the species presenting the most defined markings, i.e. serval, and compared the position of spots on individuals’ flanks [10]. The method consisted of processing all the pictures coming from one camera station, in chronological order, and then going to the next camera station. Each identified individual was characterised by an ID number, its sex (if determinable), and a distinctive spot arrangement (indicated in red in S1 Appendix pictures). The comparison of the coat markings of individuals featuring in subsequent records with identified individuals extended over the whole flank, with the key marker acting as a starting point to ease the process. We repeated the operation with striped hyaena and aardwolf, but this time focusing on fore-quarters and hind-quarters and selecting several key markers. We added a paragraph and some pictures in S1 Appendix to clarify the identification process.

Secondly, the authors note that that large carnivores preferentially use roads for movement and the camera photos used were originally collected for a study of leopards. However, there is no mention of the potential intraspecific interactions between the three focal species in this study and leopard (or even interactions among the three species). Was this considered as a covariate in the model and if so, why was it not included? Adding addition information and discussion about the intraspecific responses would be helpful.

Response:

This is an excellent suggestion, which we have in fact addressed, but the resulting set of analyses was too extensive to contain easily within a single paper. We therefore plan to submit a separate paper that looks at spatial correlates of distribution, including inter-specific interactions.

 

References:

1. Brown JH. On the relationship between abundance and distribution of species. Am Nat. 1984; 124(2): 255-279.

2. Edwards S, Portas R, Hanssen L, Beytell P, Melzheimer J, Stratford K. The spotted ghost: density and distribution of serval Leptailurus serval in Namibia. Afr J Ecol. 2018; 56: 831–840. doi: 10.1111/aje.12540.

3. Loock DJE, Williams ST, Emslie KW, Matthews WS, Swanepoel LH. High carnivore population density highlights the conservation value of industrialised sites. Sci Rep. 2018; 8: 16575. doi: 10.1038/s41598-018-34936-0.

4. Foster RJ, Harmsen BJ. A critique of density estimation from camera-trap data. J Wildl Manage. 2012; 76(2): 224–236. doi: 10.1002/jwmg.275.

5. Suryawanshi KR, Khanyari M, Sharma K, Lkhagvajav P, Mishra C. Sampling bias in snow leopard population estimation studies. Popul Ecol. 2019; 61(3): 268–276. doi: 10.1002/1438-390X.1027.

6. Doherty P, Boulinier T, Nichols J. Local extinction and turnover rates at the edge and interior of species' ranges. Ann Zool Fennici. 2003; 40(2): 145-153.

7. O'Brien TG, Kinnaird MF. Density estimation of sympatric carnivores using spatially explicit capture-recapture methods and standard trapping grid. Ecol Appl. 2011; 21: 2908-2916. doi:10.1890/10-2284.1.

8. Efford MG. Estimation of population density by spatially explicit capture–recapture analysis of data from area searches. Ecology. 2011; 92: 2202-2207. doi:10.1890/11-0332.1. 

9. Royle JA, Chandler RB, Sollmann R, Gardner B. Spatial Capture-recapture. 1st Edition. San Diego: Academic Press; 2013. 612 p.

10. Rovero F, Zimmerman F. Camera trapping for wildlife research. Exeter: Pelagic publishing; 2016. Chapter 7, Capture-recapture methods for density estimation; p. 95-142.

11. Sollmann R. A gentle introduction to camera‐trap data analysis. Afr J Ecol. 2018; 56(4): 740– 749. doi: 10.1111/aje.12557.

12. Mills G, Hofer H. Hyaenas: status survey and conservation action plan. IUCN/SSC Hyaena Specialist Group, IUCN, Gland, Switzerland. 1998. Available from: https://portals.iucn.org/library/node/7402.

13. Ramesh T, Kalle R, Downs CT. Spatiotemporal variation in resource selection of servals: Insights from a landscape under heavy land‐use transformation. J Mammal. 2016; 97(2): 554–567. doi: 10.1093/jmammal/gyv201.

14. Harmsen BJ, Foster RJ, Quigley H. Spatially explicit capture recapture density estimates: Robustness, accuracy and precision in a long-term study of jaguars (Panthera onca). PLoS One. 2020; 15(6): e0227468. doi: 10.1371/journal.pone.0227468.

15. Dupont P, Milleret C, Gimenez O, Bischof R. Population closure and the bias-precision trade-off in spatial capture-recapture. Methods Ecol Evol. 2019; 10: 661–672. doi: 10.1111/2041-210X.1315.

16. Boron V, Tzanopoulos J, Gallo J, Barragan J, Jaimes-Rodriguez L, et al. Jaguar Densities across Human-Dominated Landscapes in Colombia: The Contribution of Unprotected Areas to Long Term Conservation. PLoS One. 2016; 11(5): e0153973. doi: 10.1371/journal.pone.0153973.

17. Satter CB, Augustine BC, Harmsen BJ, Foster RJ, Sanchez EE, Wultsch C, et al. Long‐term monitoring of ocelot densities in Belize. J Wildl Manage. 2019; 83: 283-294. doi: 10.1002/jwmg.21598.

18. Johansson O, Samelius G, Wikberg E, Chapron G, Mishra C, Low M. Identification errors in camera-trap studies result in systematic population overestimation. Sci Rep. 2020; 10: 6393. doi: 10.1038/s41598-020-63367-z.

19. Bischof R, Dupont P, Milleret C, Chipperfield J, Royle, JA. Consequences of ignoring group association in spatial capture–recapture analysis. Wildlife Biol. 2020. doi: 10.2981/wlb.00649.

20. Foley C, Foley L, Lobora A, De Luca D, Msuha M, Davenport TRB et al. A Field Guide to the Larger Mammals of Tanzania. Princeton: Princeton University Press; 2014. 320 p.

21. Efford MG. Secr 3.2 - spatially explicit capture–recapture in R. 2019 [cited 2021 February 15]. 20 p. Available from: https://cran.r-project.org/src/contrib/Archive/secr/.

22. Burnham KP, Anderson DR. Multimodel inference: understanding AIC and BIC in model selection. Sociol Methods Res. 2004; 33(2): 261–304. doi: 10.1177/0049124104268644.

23. Arnold TW. Uninformative Parameters and Model Selection Using Akaike's Information Criterion. The Journal of Wildlife Management. 2010; 74: 1175-1178. doi:10.1111/j.1937-2817.2010.tb01236.x.

24. Sollmann R, Gardner B, Belant JL. How Does Spatial Study Design Influence Density Estimates from Spatial Capture-Recapture Models? PLoS ONE. 2012; 7(4): e34575. doi: 10.1371/journal.pone.0034575.

25. Tobler MW, Powell GVN, Estimating jaguar densities with camera traps: problems with current designs and recommendations for future studies. Biol Conserv. 2013; 159: 109-118. doi: 10.1016/j.biocon.2012.12.009.

26. Sun CC, Fuller AK, Royle JA. Trap Configuration and Spacing Influences Parameter Estimates in Spatial Capture-Recapture Models. PLoS ONE. 2014; 9(2): e88025. doi: 10.1371/journal.pone.0088025.

27. Clark JD. Comparing clustered sampling designs for spatially explicit estimation of population density. Popul Ecol. 2019; 61: 93–101. doi: 10.1002/1438-390X.1011.

28. Efford, MG, Boulanger, J. Fast evaluation of study designs for spatially explicit capture–recapture. Methods Ecol Evol. 2019; 10: 1529–1535. doi: 10.1111/2041-210X.13239.

---

## [Editor Report · Decision Letter 1]

17 Mar 2021

Density responses of lesser-studied carnivores to habitat and management strategies in southern Tanzania’s Ruaha-Rungwa landscape

PONE-D-20-33931R1

Dear Dr. Hardouin,

We’re pleased to inform you that your manuscript has been judged scientifically suitable for publication and will be formally accepted for publication once it meets all outstanding technical requirements.

Kind regards,

Bi-Song Yue, Ph.D

Academic Editor

PLOS ONE

---

## [Editor Report · Acceptance letter]

19 Mar 2021

PONE-D-20-33931R1 

Density responses of lesser-studied carnivores to habitat and management strategies in southern Tanzania’s Ruaha-Rungwa landscape 

Dear Dr. Hardouin:

I'm pleased to inform you that your manuscript has been deemed suitable for publication in PLOS ONE. Congratulations! Your manuscript is now with our production department. 

Kind regards, 

on behalf of

Dr. Bi-Song Yue 

Academic Editor

PLOS ONE